# Trained Immunity as a Trigger for Atherosclerotic Cardiovascular Disease—A Literature Review

**DOI:** 10.3390/jcm11123369

**Published:** 2022-06-12

**Authors:** Natalia Anna Zieleniewska, Małgorzata Kazberuk, Małgorzata Chlabicz, Andrzej Eljaszewicz, Karol Kamiński

**Affiliations:** 1Department of Population Medicine and Lifestyle Diseases Prevention, Medical University of Białystok, 15-259 Bialystok, Poland; drobeknatalia@gmail.com (N.A.Z.); mchlabicz@op.pl (M.C.); 2Department of Cardiology, Teaching University Hospital of Białystok, 15-259 Bialystok, Poland; 3Scientific Group of Department of Population Medicine and Lifestyle Diseases Prevention, Medical University of Białystok, 15-259 Bialystok, Poland; malgorzata.kazberuk@gmail.com; 4Department of Invasive Cardiology, Teaching University Hospital of Białystok, 15-259 Bialystok, Poland; 5Department of Regenerative Medicine and Immune Regulation, Medical University of Białystok, 15-259 Bialystok, Poland; andrzej.eljaszewicz@umb.edu.pl

**Keywords:** atherosclerosis, trained immunity, pathogenesis

## Abstract

Atherosclerosis remains the leading cause of cardiovascular diseases and represents a primary public health challenge. This chronic state may lead to a number of life-threatening conditions, such as myocardial infarction and stroke. Lipid metabolism alterations and inflammation remain at the forefront of the pathogenesis of atherosclerotic cardiovascular disease, but the overall mechanism is not yet fully understood. Recently, significant effects of trained immunity on atherosclerotic plaque formation and development have been reported. An increased reaction to restimulation with the same stimulator is a hallmark of the trained innate immune response. The impact of trained immunity is a prominent factor in both acute and chronic coronary syndrome, which we outline in this review.

## 1. Introduction

Atherosclerosis represents a primary public health challenge and the leading cause of cardiovascular diseases (CVD) [1]. Scientific reports on the influence of innate immune response on the development of atherosclerotic lesions have recently become widely available [2]. Notably, chronic non-resolving low-grade inflammation of the arterial wall represents a hallmark of atherosclerosis [3,4]. Recent evidence shows that innate immune response plays an important role in the disease’s initiation, progression, and final thrombotic complications [3]. The innate immune response can adopt a functional memory after a previous challenge with a stimulus, resulting in an increased (trained immunity) or decreased (immune tolerance) response towards secondary stimulation [5].

In this review, we will discuss the mechanisms of trained immunity, to atherosclerotic cardiovascular disease (ASCVD), in the context of the global pandemic.

## 2. Trained Immunity

Host immune responses are classically divided into innate and adaptive immune systems, which are often described as the opposite types of the host response. Although they differ in their mechanisms of action, the synergy between them is indispensable for a fully effective immune response [6]. The innate immune system includes physical barriers, such as layers of epithelial cells, a layer of secreted mucus that covers the epithelium in the respiratory or gastrointestinal tract, and, also, the epithelial cilia [7]. Furthermore, the innate response involves soluble proteins and small bioactive molecules that are either made of biological fluids (such as complement proteins [8] or defensins [9]) or that are released from the cells as they are activated (including cytokines or chemokines). Finally, the innate immune system contains membrane-bound receptors and cytoplasmic proteins that bind molecular patterns expressed on the surfaces of invasive microbes [7].

The innate immune response is dominated by germline-encoded pattern recognition molecules. The adaptive response is based primarily on antigen-specific receptors expressed on the surface of T and B lymphocytes [10]. These antigen-specific receptors are encoded by genes that are rearrangements of germline gene elements. They bind with intact T cell receptors and immunoglobulin B cell antigen receptors. The formation of the antigen receptors from several hundred germline-encoded gene elements allows for an infinite variety of possible antigen receptors, each of which has a potentially unique specificity for a different antigen [10]. The result is that the adaptive immune system has exceptional specificity for its target antigens.

Until a few years ago, it was thought that only an adaptive immunity could build up immunological memory [11]. However, it has been proven that innate immunity may be affected by prior exposure to the pathogens or their products, and this ability has been referred to as innate immune memory [5]. To be more precise, trained immunity represents a functional modification of the cells in the innate immune system to promote a more intense (defined as trained immunity) or less intense (defined as trained tolerance) response to the secondary challenge. Rather than creating antibodies in preparation for a second encounter with a known antigen, immunity is mediated primarily by epigenetic reprogramming and changes in gene expression and cell physiology without changes in the native DNA sequence [12].

In fact, trained immunity provides protection against secondary infection through mechanisms independent of T/B cell adaptive responses [13]. Trained immunity, in contrast to adaptive immune memory, lacks specificity and, thus, may cause a cross-reaction and protect against a different pathogen than the one it was originally exposed to [13]. The evidence of innate immune memory is mainly found in monocytes, macrophages, and the natural killers (NK) cells [6,13]. Furthermore, these mechanisms enable better pathogen recognition by pattern recognition receptors (PRRs) and result in an enhanced protective inflammatory response [5,13]. PRRs, which recognize only specific epitopes, are expressed on the majority of myocardial cells [14].

The most important receptors for this review include toll-like receptors (TLRs) and NOD-like receptors (NLRs). TLRs represent the family of transmembrane or cytosolic receptors. All TLRs are expressed in the heart [14]. The release of inflammatory cytokines, chemokines, and type I interferon (IFN) is induced by the activation of TLRs [14]. Moreover, they could recognize a wide range of DAMPs (danger-associated molecular patterns) derived from their own cells (such as RNA, DNA, and HMGB1) and PAMPs (pathogen- associated molecular patterns, including CpG, viral RNA, or LPS). The activation of other cytosolic receptors, such as NLRs, leads to the inflammatory process and the activation of interleukin (IL) 1ß, IL-18, and transcriptional activity [15]. NLRs recognize microbial products and danger signals [15].

Trained immunity, as mentioned above, is mediated by epigenetic and metabolic reprogramming. It is present in both mature cells and their bone marrow progenitors [16]. This innate immune memory is characterized by an increased cytokine response to a second challenge [5]. Importantly, trained immunity is non-specific and long-lasting—monocytes and macrophages adopt a phenotype that allows a stronger response to subsequent or unrelated stimuli, leading to chronic inflammation [17].

Trained immunity has a protective function as a host response to repeated infections, but it can also lead to chronic inflammation and, thus, to various diseases [16]. Recently, an increasing number of studies linking trained immunity to the pathogenesis of ASCVD are emerging. We presented the trained immunity mechanism on the Figure 1 (based on the article by Netea et al. [16]).

## 3. Trained Immunity May Play a Role in Atherosclerosis

The development of ASCVD is generally believed to be fueled by the persistence of low-grade vascular inflammation and the permanent hyperactivation of the innate immune system [17]. Both innate and adaptive immunity parameters of the immune response are involved in the initiation, progression, and ultimate thrombotic complications of atherosclerosis [18]. In addition to the role of macrophages and CD4+ T cells, the pathogenesis of ASCVD will be affected by regulatory T cells, dendritic cells, and mast cells [16]. After the activation of the innate immune response, an adaptive response is initiated that further promotes the formation of atherosclerotic plaque. More details on the involvement of the specific and non-specific immune responses in the pathogenesis of atherosclerosis are described in the review article by Packard et al. [18].

Recent studies suggest that the innate immune system cells, after a short stimulation with microbiological products or endogenous atherogenic stimuli, can develop a long-lasting proinflammatory phenotype [19]. Other immune and non-immune cells involved in the various stages of ASCVD can develop comparable memory specifications, as well as monocytes [16]. In this paragraph, we review the influence of trained immunity on acute and chronic ASCVD.

### 3.1. Acute Coronary Syndrome as a Consequence of Short-Term Innate Immunity Alterations

Acute coronary syndrome (ACS) is one of the clinical presentations of coronary artery disease (CAD). Chronic coronary syndromes (CCS) and ACS have different clinical consequences and distinct pathophysiology [20]. While CCS generally describes slowly progressing atherosclerotic lesions in coronary arteries, ACS is a term relating to unstable atherothrombosis, resulting in sudden symptoms of myocardial ischaemia, such as unstable angina pectoris, non–ST-segment elevation myocardial infarction (NSTEMI), and ST-segment elevation myocardial infarction (STEMI) [21].

Scientists widely agree upon an inflammatory origin of ACS, where the local immune response plays a key role. Various triggering factors influence the state of the vessel’s endothelial cells. Chronic damage of the endothelium by hyperlipidemia, shear stress, and a low-grade inflammatory state with its mediators are those which result in its dysfunction [22]. This leads to increased permeability, the adhesion of leukocytes, and an imbalance in the production and release of cell-adhesion molecules, resulting in proinflammatory cytokines and chemokines creating an inflammatory process in the vascular walls [22,23]. Physiologically, vascular endothelial cells preserve a non-adhesive, antithrombotic surface. The dysfunction of the endothelium allows the entry of oxidized and glycated lipoproteins, mainly the LDL fraction, into the vascular walls. The adherence of blood monocytes and other leukocytes to the endothelium is increased, followed by their migration to the intima and transformation into macrophages and foam cells [22]. Platelets that are activated by the tissue factor, macrophages, and vascular cells, release agents that cause smooth muscle cells to migrate from the tunica media to the intima. The proliferation of the smooth muscle cells in the intimal membrane and the production of the extracellular matrix cause alterations in the deposition of collagen and proteoglycans. An increased lipid accumulation occurs both in the cells and extracellularly [22]. Recently, cytotoxic lymphocytes have been shown to play a role in the development and progression of atherosclerotic plaques [23]. Notably, cytotoxic populations, such as invariant natural killer T cells, NK cells, γδ-T cells, CD8+ T cells, and human CD4 + CD28- T cells, are involved in the development of inflamed and unstable atheromas [23].

An atherosclerotic plaque rupture is followed by an exposition of endothelial collagen and the formation of a thrombus within an occlusion of a coronary artery, thus, being the direct cause of ACS [24]. In the aftermath, patients suffer from ischemic chest pain as a sign of restricted blood flow to the epicardium or even of total obstruction of a vessel, causing STEMI. Such patients ought to be given either a pharmacological treatment or percutaneous intervention as a reasonable approach in order to produce reperfusion and prevent necrosis from becoming transmural. On the other hand, patients with non-obstructive plaques in the coronary arteries, and enduring chest pains, are diagnosed with unstable angina pectoris (UA) or NSTEMI, depending on elevated troponin as a marker of myocardial injury.

There are several features of atherosclerotic plaques that trigger their instability and rupture. A thin layer of a fibrous cap covers a thick lipid core containing foam cells, matrix metalloproteinases, calcium, and cellular debris. The layer of smooth muscle tissue underneath the plaque is relatively diminished. By performing angiographic detection, it is possible to overlook an unstable atherosclerotic plaque, which may result in thrombosis later [25]. In this section, we will be focusing on the local and systemic inflammatory influence on atherosclerosis in the pathogenesis of ACS, the occurrence of shear stress, and the prothrombotic factors, depending on individual variability.

Several recent studies have presented new data on the inherence of an innate immune system of the heart muscle that reduces its tissue impairment as it controls its own homeostatic state. This phenomenon is founded on the disturbing balance between pro- and anti-inflammatory cells and biochemical substances. Inflammatory cytokines and chemokines that promote inflammation and contribute to the regulation of monocyte infiltration are generated by macrophages [26]. The atherosclerotic plaque contains a wide spectrum of proinflammatory cytokines, such as IL-1ß, IL-6, IL-12, IL-15, IL-18, the tumor necrosis factor (TNF), and the macrophage migration inhibitory factor (MIF) [27]. IL-1ß and IL-6 were shown to play an important role in atherosclerosis by the activation of innate immune cells, including monocytes and macrophages [27]. Elevated levels of cytokines and chemokines involved in the formation of atherosclerotic plaque, including IL6, IP-10, the macrophage inflammatory protein, 1 α and ß, and the monocyte chemoattractant protein, 1 (MCP-1), may result in an augmented inflammation and possibly a higher incidence of ACS.

IL-1ß is an essential cytokine expressed by many cells, including macrophages, NK cells, monocytes, and neutrophils. IL-1ß signaling in atherosclerosis pathogenesis includes increasing endothelial permeability and, also, the increased adhesion of lymphocytes and neutrophils to the endothelium [28]. Moreover, DNA methylation at the promoters of the IL-1ß genes was reported, and this promoter has the highest methylation status among the cytokine genes (IL-6, IL-8), creating IL-1ß, a likely mediator in the trained immunity dependent mechanisms [29]. The Canakinumab Anti-Inflammatory Thrombosis Outcomes Study confirms the prominent role of this cytokine and inflammation in the development and progression of ASCVD [28]. In this study, and among over 10,000 participants, canakinumab (the human monoclonal antibody targeted at the IL-1ß) was shown to reduce the risk of the recurrence of myocardial infarction (MI), stroke, or death from cardiovascular causes by 15% in patients with CCS or after an MI [30].

The role of reactive oxygen species (ROS) in the pathogenesis of atherosclerosis has been confirmed by many laboratories, both in experimental animals and in humans [31]. The oxidant stress is closely related to the risk factors of atherosclerosis, such as hypercholesterolemia, hypertension, DM, or smoking [32,33,34]. Furthermore, the processes associated with atherogenesis are initiated by several important enzymatic systems, including xanthine oxidase, NADPH oxidase, and nitric oxide synthase [31]. Angiotensin II has been shown to be a major stimulator of vascular reactive oxygen species production, both from the smooth arterial muscle cells and endothelial cells [35]. In addition, angiotensin II-stimulated NADPH vascular oxidase has been shown to be the main source of ROS in atherosclerosis [35]. Further data also suggest that ROS is critically involved in angiotensin II, causing a proinflammatory effect in vascular cells [36].

Proatherogenic risk factors lead to endothelial dysfunction and, consequently, to increased exposure to the adhesion proteins (e.g., selectin, the intercellular adhesion molecule, ICAM, and the vascular cell adhesion molecule-1 VCAM-1) that facilitate the activation of monocytes and their adhesion to the dysfunctional area [37]. Along with monocytes, some T cells, B cells, and NK cells may also undergo diapedesis/transmigrate into intima [38,39]. Recently, trained immunity has also been shown in NK cells, endothelial cells, and vascular smooth muscle cells [38]. Endothelial dysfunction enhances platelet activation and aggregation and increases the penetration of circulating lipids into the intimal layer [37]. Low-density lipoprotein (LDL) cholesterol undergoes oxidation when binding to proteoglycans. Oxidized lipids trigger a range of proinflammatory responses mediated by TNF-α, IL-1, and MCP-1, which further activates and recruits monocytes, macrophages, and inflammatory cells [37]. Macrophages are the ones that are associated with the progression of atherosclerosis and plaque rupture. Having been lured into the inflammatory site, monocytes tend to differentiate into an M1 type, which forms proinflammatory colonies. Also, chemokines, such as MCP-1 and CX3CL1, can recruit NK cells to the arterial wall and then trigger cytokine production [38]. This process is relevant to the formation of an unstable plaque [40]. Patients with ASCVD have increased CD14+CD16+ monocytes and serum TNF-α levels, which may indicate the possibility of a trained phenotype [41].

Many scientists agree upon the significance of scavenger receptors (SR), as well as Toll-like receptors (TLRs), appearing to have been providing a mediating role as a component of an intrinsic immunological response due to stress factors. SRs play a key role in the phagocytosis of oxidized LDL (oxLDL) by foam cells, while the Toll-like receptors are involved in plaque formation [37]. These receptors trigger an intracellular signalling cascade that stimulates the production of proinflammatory mediators and cytokines [37]. TLRs are found mainly on the surface of macrophages and endothelium, and they bind oxLDL, HSP60, lipopolysaccharides, and other ligands [42]. Moreover, this may indicate that these are receptors involved in the formation of the trained phenotype.

Furthermore, the dysfunction of endothelium enables LDL to interweave the vessel wall, resulting in its oxidation to a minimally modified LDL (mmLDL), as well as immunogenic oxLDL. Not only are these forms of lipoproteins found in atherosclerotic plaques, but they may also be found on the surfaces of cells, as well as in the circulation of patients with CAD. Within the role of innate immunity, mmLDL is internalized into atherosclerotic plaque via TLR4, resulting in the formation of foamy macrophages. TLR4s are found to be expressed more intensively in the peripheral blood mononuclear cells (PBMCs) of patients with severe CAD [43]. On the other hand, oxLDL is identified by SRs, such as CD36 and MSR-1, directing TLRs to propagate proinflammatory responses which promote vascular pathology [44]. In patients suffering from diseases involving a chronic inflammatory state with a high risk of thrombosis, oxLDL plays a meaningful role as it triggers an inflammatory process. OxLDL was found to promote the proinflammatory activation of T cells, monocytes, and endothelial cells. It also activates the cells located in atherosclerotic lesions; therefore, it incites local chronic inflammatory processes and the rupture of the plaque [44]. This is supported by the research study where hypercholesterolemic mice were proven to have elevated levels of oxLDL in their circulation, as well as excessive prothrombotic processes. Also, a higher expression of TF was found in leukocytes [44]. In the study, the authors also emphasized the role of oxLDL in inducing the TF expression depending on the TLR phenotype. Mice with hypercholesterolemia with TLR4-**/**- and TLR6-**/**- had a lesser risk of thrombotic events than those without such a phenotype. Therefore, oxLDL remains a valid risk factor for vascular inflammation and thrombosis [44].

Despite the undoubted contribution of the plasmatic factors to coagulation, the innate immunity system also responds to inflammatory processes ongoing in blood vessels. It should be stressed that chronic, or even singular, episodes of an inflammatory process generate a great risk of thrombotic events, including the pathophysiology of CAD. In fact, several studies have revealed that there is a correlation between high levels of anti-oxLDL immunoglobulin M (IgM) antibodies and a lower risk of CAD [44]. Many studies have shown that patients with no or mild CAD had increased levels of IgM in comparison with patients with severe CAD, having taken into consideration such factors as age, smoking, and LDL levels [45,46,47]. It was stated that IgM levels decrease with age and are especially low in patients with angina pectoris, hypertension, and congestive heart failure [47].

### 3.2. Effect of Trained Immunity on CCS

There are many benefits of trained immunity; however, it also plays a major role in the pathogenesis of ASCVD. The proinflammatory microenvironment in the vascular wall that promotes atherosclerotic plaque formation is formed from a complex interaction between endothelial cells, smooth muscle cells, and circulating immune cells [48]. In the following paragraph, we discuss molecules that can affect the trained phenotype and, thus, promote the CCS.

Foam cells, a type of macrophages located on the walls of blood vessels, absorb various substances involved in plaque growth [49]. In the 1980s, it was found that LDL must undergo certain structural changes to achieve atherogenic properties [50] and to be uptaken by scavenger receptors [51]. Enzymatic modifications of LDL particles include oxidation and binding to immunoglobulin complexes [52]. The formation of oxLDL is crucial for the progression of atherosclerosis. These molecules are captured by scavenger receptors, such as CD36 and MSR-1, acting in cooperation with TLRs [44]. Macrophages, which are formed upon the recruitment of monocytes to the arterial walls, are involved in LDL oxidation through enzymatic reactions and ROS secretion [53]. Then, oxidized LDLs trigger inflammatory signaling through a heterodimer of Toll-like receptors 4 and 6 [54]. A prior study showed that the activation of the CD36–TLR4–TLR6 heterotrimer stimulates the sterile low-grade inflammatory process [54]. OxLDL uptake by the macrophage and the formation of intracellular cholesterol crystals act as ligands for the NLRP3 inflammasome, leading to its activation and production of IL-1β and IL-18 [55]. Moreover, the in vitro incubation of macrophages with oxidized LDL led to the accumulation of cholesterol esters, which was not observed with native LDL [56]. Elevated plasma levels of oxidized LDL are associated with coronary artery disease [57], which was angiographically documented [58]. Furthermore, enhanced plasma levels of MDA-modified LDL suggest plaque instability [57]. It has also been shown that the atherogenic potential of lipoprotein (a) may be due, in part, to its association with oxidized phospholipid molecules [58]. Other researchers have found that short monocytes’ stimulations, with a low concentration of oxLDL, induce the proatherogenic macrophage phenotype via epigenetic histone modifications—these include increased trimethylation of lysine 4 at histone 3 in the promoter regions of TNF-α, IL-6, IL-18, MCP-1, MMP-2, MMP-9, and CD36 [59]. This modification leads to the enhanced production of the mentioned proinflammatory cytokines and foam cells’ formation [59,60]. OxLDL binds to lectin-like oxLDL receptors, LOX-1, which augment the synthesis of chemokines and cell adhesion molecules involved in the adhesion of monocytes to the endothelium [61]. Moreover, proapoptotic, prooxidant and proinflammatory pathways are created [61]. Monocytes, taking up oxLDL via scavenger receptors, release TNF-α [62]. The exposure of macrophages to oxLDL may result in their profound metabolic and epigenetic transformation. Researchers have shown that oxLDL decreases the acetylation of histones 3 and 4 and the demethylation of H3K4 (histone protein 3 at lysine 4) while simultaneously increasing H3K9 trimethylation surrounding the eNOS promoter, which is consistent with the repression of eNOS transcription in endothelial cells [63]. Glycolysis and oxidative phosphorylation are regulated in cells primed by oxLDL, and this depends on the mTOR/HIF1α signal pathway [64]. Furthermore, the simultaneous inhibition of glycolysis with the 2-deoxyglucose and mTOR pathway prevented proinflammatory phenotype formation in macrophages [65]. Further evidence of the importance of trained immunity in ASCVD pathogenesis is that the ex vivo analysis of monocytes from patients with familial hypercholesterolemia showed that, even after treatment with statins, the proinflammatory phenotype of monocytes persisted, despite a decrease in cholesterol levels [14].

Additionally, PRRs are activated by danger-associated molecular patterns, including heat shock proteins (HSP) and the High Mobility Group Box 1 protein (HMGB1). DAMPs could accumulate in atherosclerotic plaque [66]. It has been shown that HSPs are overexpressed via the activation of the heat shock transcription factor 1, particularly in endothelial cells, macrophages, and smooth muscle cells in patients at high risk of ASCVD [67]. HSPs binding to the TLR4/CD14 complex initiate an innate immune system response through the secretion of IL-6 and TNF-α [68]. Simultaneously, lymphocytes T and B produce autoantibodies and autoreactive cells against HSPs, which promotes atherosclerotic progression [67,69]. The heat shock response is mainly regulated at the transcriptional level. An association between the mitogen-activated protein kinases (MAPK) activation and the HSP expression in vascular cells in atherosclerosis has been demonstrated [70]. Three MAPKs families (ERK, JNK/SAPK, and p38 MAPK) are upregulated in the vascular cells stimulated by heat shock proteins, free radicals, oxLDL, and mechanical stress [71,72,73,74]. Most of the signals are not fully understood, and thorough studies are needed on the effect of HSP on the proinflammatory phenotype. HMGB1 is a proximal trigger that is sufficient to induce the release of other cytokines associated with the mediation inflammatory response (i.e., TNF-α, IL-6) [75]. This non-histone chromosomal protein binds to DNA in a sequence-independent manner and modifies the DNA structure [75]. Moreover, it is actively released by monocytes/macrophages in response to inflammatory stimuli; then, it binds to TLR-2 or 4 and activates proinflammatory signaling pathways [76]. High levels of extracellular HMGB1 have been detected in human atherosclerosis plaque as a result of inflammation response, tissue damage, and apoptosis [77]. Moreover, it has been demonstrated that HMGB1 increases IL-1 β production in vascular smooth muscle cells [78]. Interestingly, an increase in IL-6 and TNF-α production upon HMGB1 stimulation was observed in in vitro trained immunity models [67]. HMGB1 was shown to bind at the semaphoring 3A genomic locus, promote heterochromatin formation, and decrease the occupancy of acetylated histones, which represent evidence of epigenetic changes [79].

Recently, monocytes M1 and the M2-phenotype have been shown to use different metabolic pathways during activation [80,81]. In summary, the M1-phenotype macrophages are activated by LPS and IFN-gamma and mainly produce proinflammatory cytokines in contrast to the M2-phenotype, which is stimulated by IL-4-related cytokines and expresses genes involved in tissue repair [82]. Saeed et al. documented the significance of epigenetic regulation underlying the monocyte to macrophage differentiation and trained immunity [83]. The epigenetic profiling indicated dynamic regions associated with hypersensitivity to DNase I [83]. Alterations in histone acetylation detected most of the dynamic events, and these regions demonstrated some degree of DNase I availability that was already present in monocytes.

## 4. The Risk Factors of ASCVD and Trained Mechanism

Stress, both acute and chronic, is linked with an elevated risk of ASCVD and even with the acceleration of atherosclerotic lesions in animal models [84]. Recently, researchers showed that human primary monocytes exposed to an appropriate concentration of epinephrine/norepinephrine demonstrated increased levels of TNF-α and IL-6 after restimulation of lipopolysaccharides 6 days later. This trained phenotype, via the β-adrenoceptor, was associated with expanded glycolytic capacity and oxidative phosphorylation [85]. Additionally, in a cohort of patients with pheochromocytoma, they showed a trend toward higher ex vivo TNF-α production upon LPS stimulation. Interestingly, in this group of patients, the composition of blood cells was partially normalized 4 weeks after the operation, while production of intermediate monocytes and TNF-α remained high, which may also suggest a trained mechanism [85].

Under insulin resistance, peripheral macrophages acquire an anti-inflammatory M2-like phenotype characterized by the increased expression of M2 markers and a decreased secretion of proinflammatory factors after the stimulation [86]. Likewise, insulin-resistant macrophages produce reduced levels of NO after TLR stimulation in vivo and in cultures and exhibit a reduced bactericidal capacity in vivo [86]. M2-polarized macrophages have also been found in other conditions associated with systemic insulin resistance—for example, cancer and CVD [87,88]. Recently, the inhibition of insulin signaling has been shown to be a promoter of the M2-like phenotype in peripheral macrophages [89].

Obesity induces metabolic inflammation, a chronic low-grade inflammatory condition characterized by elevated systemic levels of proinflammatory markers and insulin resistance. Obesity can affect innate immunity through the dysregulation of NK cells, an increased number of marrow cells in the adipose tissue, and the expression of ACE2 by adipocytes [90]. In animal models, genetic deletion of the insulin receptor in macrophages protects against diet-induced obesity [91]. In obesity and DM, hyperglycemia can be characterized by the persistent activation of the NF-kB gene through increased H3K4 methylation and decreased H3K9 methylation [92]. In the context of these studies, high glucose levels induce epigenetic changes and macrophage training. In obesity, the number of known trained immunity inducers is increased, which include cytokines, lipopolysaccharide (LPS), and fatty acids [93]. Bekkering et al. have proposed a mechanism for trained immunity in obesity that includes training hematopoietic stem cells (HSCs) in the bone marrow leading to the production of activated monocytes [93]. Furthermore, NK cells could be trained by LPS and cytokines, as well as adipose tissue, which can additionally be trained by fatty acids and adipokines. The described changes result in activated tissue macrophages and training of non-immune cells [93]. Further studies are necessary to fully characterize the impact of trained immunity on ASCVD risk factors.

## 5. Therapeutic Targeting of Trained Immunity

Notably, modified LDL levels may be used for the identification of patients with acute coronary syndrome [55]. Statins are commonly used in the primary and secondary prevention of ASCVD to lower the LDL cholesterol fraction [92]. A study involving oxLDL levels and CAD severity found that statin-treated patients had significantly lower oxLDL levels than untreated patients [93]. A recent review summarized the pleiotropic effects of statins, and some of these, such as the production of proinflammatory cytokines and ROS, altering endothelial nitric oxide synthase expression, and reduction platelet reactivity, may be associated with trained immunity [94]. Moreover, another study on experimental animal models has demonstrated that the simvastatin treatment reduced the prothrombotic phenotype and appeared to be the only statin to reduce oxLDL levels without affecting LDL levels [41]. However, treatment with statins did not affect the proinflammatory phenotype of monocytes in patients with familial hypercholesterolemia [95]. Moreover, another study concluded that habitual diet quality was correlated with differential peripheral leukocyte DNA methylation levels, which occur at 30 cytosine–guanine dinucleotides [96,97]. In addition, these epigenetic changes have been associated with multiple cardiovascular risk factors, such as body mass index, triglycerides, high-density lipoprotein cholesterol concentrations, and type 2 diabetes mellitus [96,97,98]. These results suggest that the genomic analysis of nutritional information may reveal molecular targets for the prevention and treatment of ASCVD.

Proprotein convertase subtilisin/kexin type 9 (PCSK-9) inhibitors are the group of drugs that, besides lowering LDL concentrations, are proven to have numerous pleiotropic effects, including an anti-atherosclerotic effect, the stabilisation of atherosclerotic plaque, and the ability to affect the course of bacterial infections [99]. As previously mentioned, decreasing active IL-1β levels with canakinumab reduced the rate of cardiovascular events in the group of patients taking this drug [28]. Although PCSK-9 inhibitors do not significantly reduce C-reactive protein levels [100], they have an effect on inhibiting ongoing inflammation by reducing IL-1α, TNF-α, and IL-6 levels [101]. Another reported mechanism is also an increase in the anti-inflammatory IL-10 [102]. The process of PCSK-9 inhibition also affects atherosclerotic plaque formation through changes in the C-C chemokine receptor type 2 expression [103]. These appealing data require further studies. Immunomodulation appears to be one of the most important therapies that can be used to influence trained immunity. We may be able to prevent the induction of trained immunity by modulating the pathways and inactivating the molecules that have been described in previous chapters.

## 6. Conclusions

There is only limited evidence that comes from research directly linking trained immunity to ASCVD. Our literature review highlights the aspects of knowledge that has the potential to advance our understanding of trained immunity and the pathogenesis of atherosclerosis. Although it is widely accepted that atherosclerosis is associated with chronic low-grade sterile inflammatory processes of the vascular wall, the precise mechanisms remain elusive. Further studies investigating the effect of epigenetic changes on the modification of innate immune cells in the context of atherosclerosis are relevant. A better understanding of those mechanisms is essential for ASCVD prevention and for creating novel therapeutic goals, which would be used in primary cardiovascular prevention.

## Figures and Tables

**Figure 1 jcm-11-03369-f001:**
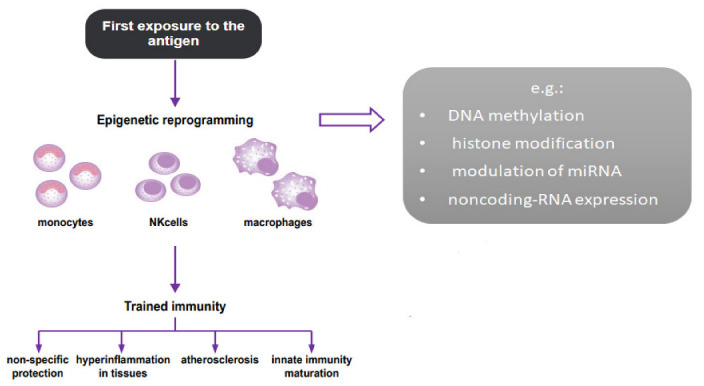
The trained immunity mechanism. The figure was created based on an article by Netea et al. [16].

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
