# Peer review of "Trained Immunity as a Trigger for Atherosclerotic Cardiovascular Disease—A Literature Review"

_jcm, 2022, doi:10.3390/jcm11123369_

Round 1

Reviewer 1 Report

This review covers an aspect of innate immunity in atherosclerosis. Innate immunity is a broad topic and the emphasis of the review is on certain areas which is valid. However, it leaves out areas which have received substantial investigation at least in animal models and which have been shown to substantially impact atherogenesis. These are specifically NKT and gamma-delta T cells, both of which are found in atheromas. These cells are not even listed on page 2. As both of these innate T cell effectors have been shown to develop a memory phenotype and may meet the authors definition of "trained immunity", it may be reasonable to at least recognize their existence.

The discussion of cytokines, chemokines, ROS, and innate effectors such as M1/M2 monocytes is reasonable but this is a complex aspect of the total response since it is very difficult to separate a specifically innate response from the adaptive immune response and the impact the adaptive immunity then has on the innate system. The adaptive immunity could still be the primary driving force although the immediate effector molecules are derived from both innate and adaptive sources.

Bringing in the discussion of TLR into CVD is good. Innate T cells, especially gamma-delta T cells are known to recognize HSP type antigens and might be another connection between innate immunity and CVD.

Overall, the review is informative and reasonable.

Minor points:

1. "Antigen" is misspelled in Figure 1.

Reviewer 2 Report

Natalia Anna Zieleniewska presented a manuscript titled: “Trained immunity as a trigger for atherosclerotic cardiovascular disease - a systematic review.” Even though the topic is interesting, there are some ambiguities that need to be addressed.

1.     First of all, this is not a systematic review. It is a popular misconception that systematic reviews have to include a meta-analysis. That is not true because there are several subtypes of systematic reviews with different analytical methodologies. However, this manuscript/study doesn’t have the characteristics of any systematic review type. It is written in a narrative style as an overview of the previously published works on the chosen topic. Hence, this is a literature/narrative review. Pleaser revise this in your manuscript.

2.     CVDs are the leading cause of death worldwide. Atherosclerosis is surely the main driver in most CVDs, hence the indirect cause, but please refrain from making such statements.

3.     Line 93-94. You should add a brief elaboration of the trained immunity pathophysiology that leads to the chronic inflammation state.

4.     You should further expand your subsection 3.1. with hypothesis and conclusions from this study (doi: https://doi.org/10.3390/biom10111514).

5.     You should expand your Conclusions with a future perspective from your point of view.

Reviewer 3 Report

Dear Authors,

I am impressed with the good review.

Some minor points:

  1. Please study the abbreviations carefully and expand them the first time when appear. And no longer in the next ones.

2.       Do the authors have permission to use the figure 1 ? (Netea MG, Joosten LA, Latz E, Mills KHG, Natoli G, Stunnenberg HG, O'Neill LAJ, Xavier RJ. Trained immunity: a program of 428 innate immune memory in health and disease. Science. 2016; 352:aaf109)?  If so, please share it. If not, please modify the figure and use the term in your own modification for [..]…

3.       In section 3. I propose to also add a part about vulnerable atherosclerotic plaque and the concentration of various cytokines and interleukins. Recent publications on this topic have appeared, e.g. in the Medicina journal, MDPI.

4.       Section 5 lacks discussion of possible pleiotropic effects of PCSK9 inhibitors as a new treatment option and recent work on the effects on pro-inflammatory cytokines. Many reviews (e.g. Molecules, IJMS).

Kind regards,

Reviewer
